# A Paradoxical Role for Regulatory T Cells in the Tumor Microenvironment of Pancreatic Cancer

**DOI:** 10.3390/cancers14163862

**Published:** 2022-08-10

**Authors:** Thomas Brouwer, Marieke Ijsselsteijn, Jan Oosting, Dina Ruano, Manon van der Ploeg, Frederike Dijk, Bert Bonsing, Arantza Fariña, Hans Morreau, Alexander Vahrmeijer, Noel de Miranda

**Affiliations:** 1Department of Surgery, Leiden University Medical Center, Albinusdreef 2, 2333 ZA Leiden, The Netherlands; 2Department of Pathology, Leiden University Medical Center, Albinusdreef 2, 2333 ZA Leiden, The Netherlands; 3Department of Pathology, Amsterdam University Medical Centre, Meibergdreef 9, 1105 AZ Amsterdam, The Netherlands

**Keywords:** pancreatic ductal adenocarcinoma (PDAC), regulatory T cells (Treg), tumor infiltrating lymphocytes (TIL), tumor microenvironment (TME), prognosis

## Abstract

**Simple Summary:**

Pancreatic cancer is one of the most lethal cancer types and its high refractoriness to therapies, including immunotherapy, has often been associated with the predominantly immune suppressive tumor microenvironment that characterizes pancreatic tumors. Regulatory T cells (Tregs) are generally considered as drivers of immune suppression in cancers. However, an increasing number of reports suggest a paradoxical association between tumor infiltration by Tregs and improved patient prognosis, in particular in gastrointestinal cancers. Here we show that Treg infiltration in pancreatic ductal adenocarcinomas (PDAC) is associated with better overall survival of patients.

**Abstract:**

Pancreatic ductal adenocarcinoma (PDAC) is considered to be a poorly immunogenic cancer type that combines a low mutation burden with a strong immunosuppressive tumor microenvironment. Regulatory T cells (Tregs) are major drivers of immune suppression but their prognostic role, particularly in gastrointestinal malignancies, remains controversial. Lymphocytic infiltration in 122 PDAC samples was assessed by multispectral immunofluorescence with anti-Keratin, -CD3, -CD8, -FOXP3 and -CD163 antibodies. Differential infiltration by Tregs was analyzed in the context of transcriptomic profiles that were available for 65 tumors. High infiltration of CD3^+^CD8^−^ (mainly CD4^+^) T cells and, especially, of the subset expressing FOXP3 (Tregs) was associated with improved patient survival, whilst cytotoxic CD3^+^CD8^+^ T cell infiltration did not have an impact on overall survival. Transcriptomic analysis revealed three signatures in PDAC tumors comprising of epithelial-mesenchymal transition (EMT)/stromal, metabolic, and secretory/pancreatic signature. However, none of these signatures explained differences in Treg infiltration. We show that Tregs associate with improved overall survival in PDAC patients. This effect was independent of cytotoxic T cell infiltration and the transcriptomic profiles of their respective tumors. These findings provide a new layer of complexity in the study of PDAC tumor microenvironment that must be considered when developing immunotherapeutic interventions for this disease.

## 1. Introduction

The immune contexture of cancers has profound implications for patient prognosis and treatment, immunotherapeutic strategies in particular [1,2]. To date, immunotherapy has made a huge impact for the treatment of previously incurable cancers with the development of T cell checkpoint blockade therapies [3,4,5]. These treatments have been most beneficial in cancer types with an immunogenic character–generally provided by a high mutation burden-through the rescue and reinvigoration of naturally-occurring anti-tumor immune responses [6,7,8]. Unfortunately, pancreatic ductal adenocarcinoma (PDAC), with the exception of mismatch repair deficient PDACs which occur in less than one percent of patients, is traditionally classified as poorly immunogenic and, therefore, patients have seen little to no clinical benefit from immunotherapeutic approaches [9,10,11,12,13]. In addition to its low mutation burden, PDAC is characterized by an abundant stromal compartment composed of immunosuppressive immune cells, fibroblasts, and extracellular matrix [14,15,16]. These features are thought to be important contributors to the dismal 5-year survival rate of PDAC patients, which is around 9% [17].

Paradigmatic driving forces supporting an immunosuppressive tumor microenvironment and typically associated with worse patient prognosis include FOXP3^+^ regulatory T cells (Tregs), tumor-associated macrophages or myeloid-derived suppressor cells [18,19,20,21,22]. In opposition, in most cancer types, the presence of cytotoxic CD8^+^ T cells (CTLs) has a predominantly positive effect on patient survival. However, in PDAC, this relation is still unclear as attested by a number of contradictory reports [22,23,24]. During tumorigenesis, the anti-cancer activity of CTLs is often negated by Tregs through a myriad of mechanisms that include the secretion of immune suppressive molecules like TGF-β, IL-10 or adenosine [25,26,27,28]. As such, Treg infiltration in tumors has mostly been interpreted as a poor predictor of tumor behavior but conflicting reports have been published on its association with clinical outcomes [1]. In fact, an increasing number of studies now associate Treg infiltration in tumors with improved clinical outcomes, especially in colorectal cancer [29,30,31,32,33]. Treg infiltration can be interpreted as a normal consequence of a competent inflammatory and cytotoxic response, thereby linking Tregs with improved survival [34]. Others have suggested that distinct Treg subpopulations associate differently with clinical outcome, thereby explaining the previous conflicting reports [35]. A recent study demonstrated a protective role for Tregs in pre-clinical PDAC models as Treg depletion failed to relieve immunosuppression in PDAC tumors and, instead, led to accelerated tumor progression [36]. These data strongly support the need to revisit the roles that cytotoxic- and regulatory T cells play in the tumor microenvironment of human PDAC in light of the development of future (immuno-) therapeutic approaches. To this end, we evaluated lymphocytic infiltration in a cohort of PDAC tissues by multispectral immunofluorescence imaging. Furthermore, we investigated a potential association between transcriptomic signatures and immune cell infiltration. Our data supports the notion of a paradoxical role of regulatory T cells in PDAC as their infiltration associates with improved clinical outcome.

## 2. Materials and Methods

### 2.1. Patient Material

Formalin-fixed, paraffin-embedded (FFPE) tumor tissues from 122 patients that underwent surgical resection between 2003 and 2015 were investigated by immunohistochemical and multispectral immunofluorescence analysis. Furthermore, snap-frozen tumor tissues from 65 of those patients were employed for transcriptome sequencing. Histopathological features were evaluated by specialized gastro-intestinal pathologists (AF and HM) whilst clinical data was obtained under an institutional review board (IRB)-approved protocol (protocol B18.049) at the Leiden University Medical Centre (LUMC). Patient samples were anonymized and handled according to the medical ethical guidelines described in the Code of Conduct for Proper Secondary Use of Human Tissue of the Dutch Federation of Biomedical Scientific Societies.

### 2.2. Immunofluorescence Procedures

For multispectral immunofluorescence (IF), analyses were performed on 4-μm FFPE tissue sections and processed as described previously [37]. In brief, FFPE tissue sections were deparaffinized with xylene and washed in ethanol. Heat-induced antigen retrieval in citrate buffer (10 mM, pH 6) was performed and the slides were allowed to cool down to room temperature. Subsequently, the tissues were blocked with Superblock buffer (Thermo Fisher Scientific, Waltham, MA, USA) and incubated overnight at 4 degrees Celsius with all primary antibodies to be detected indirectly: anti-CD8 (1:50 dilution, 4B11, DAKO technologies, Agilent Technologies, Santa Clara, CA, USA) and anti-FOXP3 (1:25 dilution, 236A/E7, Thermo Fisher Scientific). The following day, slides were washed in PBS and incubated with fluorescently-labelled secondary antibodies for one hour: CF555-labelled goat-anti-mouse IgG2b and CF633-labelled goat-anti-mouse IgG1 (both antibodies at a 1:400 dilution, Sigma–Aldrich, Saint Louis, MO, USA). After washing, directly labeled antibodies were applied: anti-pan-cytokeratin (1:50 dilution, AE1/AE3 and C11, Thermo Fisher Scientific and Cell Signaling Technology, respectively) labelled with Alexa Fluor 488 and CD3 (1:50 dilution, D7A6E, Cell Signaling Technology) labelled with Alexa Fluor 594 (Thermo Fisher Scientific). To demonstrate that the majority of CD3^+^CD8^−^ cells corresponded to CD4^+^ T cells, a similar methodology was applied but the anti-FOXP3 antibody was replaced by an anti-CD4 antibody (1:100 dilution, EPR6855, Abcam, Cambridge, UK), detected with an -Alexa Fluor 488-labelled anti-rabbit antibody (Thermo Fisher Scientific). For the separate analysis of CD163^+^ myeloid cells, the same protocol was used with an anti-CD163 antibody (1:10 dilution, 10D6, Thermo Fisher Scientific) and a CF633-labelled goat-anti-mouse IgG1 secondary antibody (Sigma–Aldrich). All immunofluorescence immunodetections were performed in the presence of a negative control where the primary antibodies were omitted. Furthermore, the immunofluorescence procedures were tested in the presence of isotype controls.

### 2.3. Image Acquisition and Cell Counting

Tissue slides were imaged at 20× magnification and, at least, three regions of interest containing cancer cells were analyzed per sample (1.34 mm^2^/region of interest) with the Vectra 3.0 Automated Quantitative Pathology Imaging System (Perkin Elmer). An analysis algorithm was trained for tissue and cell segmentation as well as immunophenotyping of cells. DAPI and keratin were employed for segmenting images into tumour, stroma, and ‘no tissue’ areas. Next, cellular segmentation was performed using a counterstain-based approach with DAPI to segment nuclei and membrane markers (CD8, CD3, CD163) to detect cell contours. The following phenotypes were identified: CD3^+^CD8^−^FOXP3^−^ T cells (which were mainly comprised of CD4^+^ T helper cells), CD3^+^CD8^−^FOXP3^+^ T cells (corresponding to Tregs) CD3^+^CD8^+^ T cells (corresponding to CTLs), and CD163^+^ myeloid cells to pinpoint tumor-associated macrophages. All images were visually inspected to confirm the correct attribution and quantification of phenotypes. For each case, cell counts were normalized by tissue area (number of cells/mm^2^) [37].

### 2.4. RNA Sequencing Analysis

For RNA isolation, 30 sections of 20 µm were cut, and RNA was isolated using RNABee (Bio-Connect, Huissen, the Netherlands) and the RNeasy Mini kit (Qiagen, Hilden, Germany) according to manufacturer’s instructions. RNA was amplified using the Total Prep RNA Amplification kit (Illumina, San Diego, CA, USA). Poly-A enriched libraries were synthesized using TruSeq RNA Library Prep kit and sequenced in three batches (Illumina HiSeq2500). All sequencing data were quality-controlled using FastQC and found to be of high quality. RNA sequencing reads were first aligned to the human reference genome (build hg38) using STAR (version 2.7.3a) [38]. The mapped RNA reads were then assigned a specific transcript/gene by the HTSeq-count tool (version 0.9.1) [39] using the GENCODE human reference gene set (v30) as reference. Limma-voom normalized counts for a list of immunomodulator genes were extracted and plotted in a heatmap using the R package heatmap.plus. The discovery of clusters defined by transcriptomic signatures was performed as described by Bailey et al. [40]. Genes from non-normalized data without at least 1 c.p.m. in 20% of the samples were excluded from further analysis. Non-negative matrix factorization (NMF) was employed to identify stable sample clusters. The preferred clustering result was determined using the observed cophenetic correlation between clusters and the average silhouette width of the consensus membership matrix as determined by the R package ‘cluster’. The R package ‘ConsensusClusterPlus’ was employed to verify sample clustering and to subtype PC samples according to the expression signatures defined in Moffitt et al. [41].

### 2.5. Statistical Analyses

Cell counts are presented as median with interquartile range (IQR). Overall survival analyses were performed by using Kaplan–Meier plots and log-rank (Mantel–Cox) tests in SPSS (version 24.0.0). The correlation between the presence of different immune cell subsets was tested by applying Spearman correlation analyses with Graphpad (version 8.4.2).

## 3. Results

### 3.1. The Lymphocytic Infiltrate of PDAC Is Dominated by T Helper Cells

To investigate the relative abundance of different T cell subsets within the tumor microenvironment of PDAC we performed immunofluorescence analysis by multispectral fluorescence imaging. We simultaneously assessed the infiltration of cytotoxic T cells (CD3^+^CD8^+^), T helper cells (CD3^+^CD8^−^FOXP3^−^) and Tregs (CD3^+^CD8^−^FOXP3^+^) in the tumor microenvironment of 122 treatment-naïve PDAC samples (Figure 1a). Most PDAC cases were characterized by an abundant stromal compartment and poorly infiltrated by T cells; cytotoxic CD3^+^CD8^+^ T cells in particular (Figure 1a,c). Most immune cells were found in the stromal compartment of tumors with only very little cells demonstrating intraepithelial localization (Appendix A). Four cases displayed more than 200 cytotoxic CD3^+^CD8^+^ T cells per area of tissue (mm^2^) (Figure 1b), of which two were long-term survivors (>5 years, in total 17 patients within the cohort were long term survivors), indicating potentially immunogenic tumors. Indeed, one of the PDAC samples with the highest count of cytotoxic CD3^+^CD8^+^ T cells was determined to be the only DNA mismatch repair-deficient PDAC case in this cohort, as determined by loss of MSH2 and MSH6 protein expression (data not shown). The CD3^+^CD8^−^ population was confirmed to be mainly comprised of CD4^+^ T cells (Appendix A). These were the most abundant subset, and, on average, 34% (5–81%) of these cells comprised Tregs, as deduced from FOXP3 nuclear expression (Figure 1b, cyan). The abundance of CD3^+^CD8^−^(CD4^+^) T cells (including Tregs) was strongly correlated with the one of cytotoxic CD3^+^CD8^+^ T cells and Tregs (*r* = 0.5811, (*p* < 0.0001) and 0.8089, (*p* < 0.0001), respectively, Spearman correlation) (Figure 1d,e). In contrast, infiltration by Tregs was only moderately correlated to cytotoxic CD3^+^CD8^+^ T cell infiltration (*r* = 0.4510 (*p* < 0.0001), Spearman correlation) (Figure 1f). Interestingly, FOXP3 expression was also detected in cancer cells, in 20% of the tumors. This phenomenon was previously reported and associated with high levels of TGF-β activation in cancer cells (Appendix A) [42].

### 3.2. Treg Infiltration Is Associated with Improved Prognosis in PDAC

To determine if T cell infiltration in PDAC was associated with overall survival, samples were divided into low or high infiltration groups based on the median counts for each T cell population. Higher infiltration by total CD3^+^CD8^−^(CD4^+^) T cells (including Tregs) within the PDAC tumor microenvironment was associated with longer overall patient survival (Figure 2a, *p* = 0.001, Log rank Mantel–Cox test). Within this population, a higher infiltration by Tregs (CD3^+^CD8^−^FOXP3^+^) also associated with improved survival (Figure 2b, *p* = 0.009, Log rank Mantel-Cox test) but no significant association was found for the CD3^+^CD8^−^FOXP3^−^ subset, likely to be comprised of T helper cells (Figure 2c). Strikingly, no association was also found between differential infiltration of cytotoxic CD3^+^CD8^+^ T cells and PDAC patient overall survival (Figure 2d).

### 3.3. Treg Infiltration Is Not Associated with Specific Transcriptional Profiles in PDAC

In line with our findings, Treg depletion was recently reported to accelerate pancreatic carcinogenesis [36]. To investigate in the current cohort whether Treg infiltration was associated with other biological features, we compared gene expression profiles between Treg-high and Treg-low tumors. By interrogating gene sets comprising of immune-related genes (Appendix A), as well as fibroblast-associated genes and TGF-β signaling targets (Appendix A), we could not identify a transcriptomic signature associated with Treg infiltration.

We then investigated whether molecular subtypes could be defined in our cohort based on gene expression signatures, as described previously [40]. Three well-defined clusters could be identified that grouped samples based on a (1) epithelial-to-mesenchymal and stromal signature, a (2) metabolic signature, and a (3) pancreatic, secretory signature driven by the expression of genes associated to insulin production (Appendix A). Contrary to what was previously reported, we could not discern a group of samples with an immunogenic subtype. Furthermore, none of these molecular subtypes associated with patient survival and Treg infiltration (Appendix A) [43,44,45].

Subsequently, we set out to determine a potential association between Treg infiltration and the presence of myeloid cells with an immune suppressive phenotype, as compensational myeloid derived immune suppression has been shown by Zhang et al. to be increased after Treg depletion [36]. To achieve this, we evaluated the number of CD163^+^ tumor-associated macrophages in our cohort [46,47]. No increase of tumor-associated macrophages was found in the group with low Treg infiltration. Moreover, no correlation was found between the number of Tregs and tumor-associated macrophages across the cohort (Spearman *r* = −0.1190 (*p >* 0.05)) (Appendix A).

## 4. Discussion

PDAC remains as one of the most complex cancer types to study from an immunological point of view. The stromal compartment in PDAC plays a major role in the pathogenesis of this disease. It is, in general, largely deprived of immune cells and those that are present often have pro-tumorigenic features. The failure of immunotherapy in PDAC is often attributed to its non-immunogenic properties that are validated by the lack of naturally-occurring anti-tumor immune responses, in particular, cytotoxic T cell-mediated activity. In line with expectations, the amount of cytotoxic CD3^+^CD8^+^ T cell infiltrate within this cohort was low. Furthermore, cytotoxic CD3^+^CD8^+^ T cell infiltration showed no association with survival, although two out of the four PDAC patients with the highest cytotoxic CD3^+^CD8^+^ T cell infiltrate were long-term survivors. CD3^+^CD8^−^(CD4^+^) T cells, including Tregs, were the most abundant T cell population within this PDAC cohort. Prior research has indicated that PDAC’s distinctive immunosuppressive microenvironment is largely driven by Tregs [22,48,49,50]. However, their functional role in the immune contexture of cancer has been increasingly described as ambiguous, also in PDAC [51,52]. Recent preclinical findings show that depletion of Tregs fails to thwart tumor growth. By making use of pancreatic cancer mouse models, Zhang and colleagues demonstrated that the depletion of Tregs resulted in the remodeling of the tumor microenvironment as a result of the decreased availability of TGF-β ligands as well as the upregulation of several chemokines involved in the recruitment of myeloid cells. This translated to the replacement of fibroblast subsets and the influx of myeloid cells with immunosuppressive features [36]. In line with these findings, we show that increased infiltration by Tregs associates with longer overall patient survival in PDAC. Carstens et al., performed a similar investigation to the one reported here but did not find a significant association between Treg infiltration and patient prognosis and, instead, reported an association between cytotoxic T cell infiltration and PDAC survival [53]. A major difference between our studies that might contribute to the distinct outcomes is that Carstens and colleagues performed their observations on tissue microarrays, while in this study we employed full tissue sections.

It has been proposed that divergent reports regarding the prognostic role of Tregs in cancer can be partly explained by the existence of different Treg subsets. Saito and colleagues demonstrated that, in colorectal cancer, two different Treg subsets can be distinguished by different levels of expression of FOXP3 [35]. Furthermore, they proposed that those subsets contributed in opposing ways to colorectal cancer outcomes with the FOXP3-low subset associating with improved patient survival. To acquire mechanistic insight into our paradoxical findings in regard to the potential protective role of Tregs, we investigated whether tumor transcriptional profiles associated to the infiltration of this subset. However, no association was found with immune-related or stromal and TGF-β -related signatures. Using a previously proposed strategy to stratify PDAC according to molecular features, we defined three subgroups in our cohort defined by an EMT/TGF-β, metabolic, and pancreatic signatures [40]. Interestingly, and contrary to what was previously reported by Bailey et al. we did not identify an immunogenic subtype, characterized by upregulated immune networks and T- and B cell infiltration [40]. We hypothesize that differences between cohorts might underly these findings.

A limitation of our study was that we only investigated the presence of three lymphocyte subsets and could not discern additional populations. Going forward, more detailed studies that allow the identification of different subpopulations in immune cell subsets in pancreatic cancer are required, also to determine whether different Treg subsets translate to distinct clinical outcomes. On the other hand, the fact that immune-related features do associate with the clinical behavior of pancreatic cancer hints towards the possibility of immunomodulatory therapies being successful in the future for the treatment of this disease.

## 5. Conclusions

Our work suggests a protective role of Tregs within the pancreatic cancer tumor microenvironment. It remains unclear what is the driving force supporting Treg infiltration in these tumors, and, more importantly, how Tregs might impact patient prognosis. Additional studies will be required to decipher the role of Tregs in PDAC and, eventually, to demonstrate how the observations reported here can translate to the clinical management of patients.

## Figures and Tables

**Figure 1 cancers-14-03862-f001:**
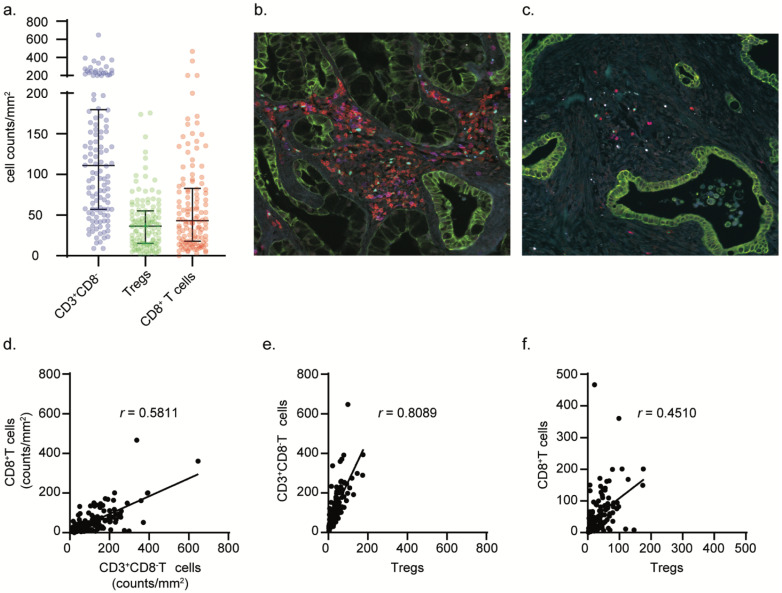
PDAC samples display heterogeneous amounts of T cell infiltration with CD3^+^CD8^−^(CD4^+^) T cells (including Tregs) being the most abundant population. Distribution and frequencies of analyzed phenotypes in PDAC samples (**a**). Representative images of the PDAC tumor microenvironment displaying high- and low lymphocytic infiltrate (**b**,**c**). Correlation analyses between cytotoxic CD3^+^CD8^+^ T cell and CD3^+^CD8^−^(CD4^+^) T cell counts (**d**), CD3^+^CD8^−^(CD4^+^) T cell and Treg counts (**e**), and cytotoxic CD3^+^CD8^+^ T cell and Treg counts (**f**). Spearman correlation coefficients (*r*) are presented per analysis.

**Figure 2 cancers-14-03862-f002:**
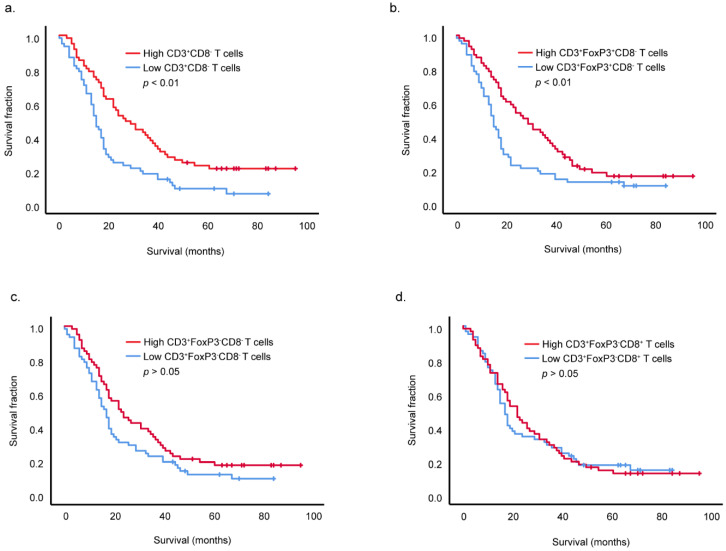
CD3^+^CD8^−^(CD4^+^) T cells (including Tregs) and Treg infiltration positively associate with improved overall survival of PDAC patients. Survival analyses of all 122 PDAC patients based on high and low infiltration by CD3^+^CD8^−^(CD4^+^) T cells (including Tregs) (**a**), CD3^+^CD8^−^FOXP3^+^ Tregs (**b**), CD3^+^CD8^−^FOXP3^−^ cells (CD4^+^ T helper cells) (**c**), cytotoxic CD3^+^CD8^+^ T cells (**d**). High and low infiltrated groups were formed on the basis of median counts. Significance was determined by using the Log rank Mantel-Cox test.

## Data Availability

Data are available upon reasonable request.

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
