# Peer review of "A Paradoxical Role for Regulatory T Cells in the Tumor Microenvironment of Pancreatic Cancer"

_cancers, 2022, doi:10.3390/cancers14163862_

Round 1

Reviewer 1 Report

The article entitled: A paradoxical role for regulatory T cells in the tumor microen- 2 vironment of pancreatic cancer; by Brouwer et al., is overall well written, however; it needs some amendments to improve the manuscript before publication:
Majors:
- Suplementary figures must exhibit the error bars and  statistical analyses. Specially:  Suppl Fig 3b,  6b and 7a.
Minors:
-Introduction describes role of some T cell population; however, it lacks information related to markers associated to the population you identified in the tumors e.g.: CD3, CD8, FOXP3, and CD163.
-Statistical analysis is poorly described. Please supplement with the specific tests used to evaluate the results. Justify Pearson tests used for linear correlations.
-Please avoid discussion in the results section since you have already a discussion section.
-Which kind of survivals have you analysed? PFS or OS?

Author Response

The article entitled: A paradoxical role for regulatory T cells in the tumor microenvironment of pancreatic cancer; by Brouwer et al., is overall well written, however; it needs some amendments to improve the manuscript before publication:

We thank the Reviewer for their comments and provide a point-by-point answer below. Of note, most of those have already been addressed in the previous round of review and were specifically commented on the point-by-point answer included in the cover letter. We wonder whether this has been made accessible to the Reviewer.

Majors:

- Suplementary figures must exhibit the error bars and  statistical analyses. Specially:  Suppl Fig 3b,  6b and 7a.

Figures 3b and 6b represent sample numbers, not measurements and, therefore, no error bars can be presented. In supplemental figure 7A, medians with interquartile ranges are shown which is the appropriate representation for this type of data. No statistical analyses has been performed as no cues for major differences are provided by graphical comparison. The group’s practices regarding statistical analyses avoid repeated use of statistical measures to investigate differences unless those are clearly apparent from graphical representation (in line with advice from statistical support).

Minors:

-Introduction describes role of some T cell population; however, it lacks information related to markers associated to the population you identified in the tumors e.g.: CD3, CD8, FOXP3, and CD163.

We find it more appropriate to introduce these markers in the Results section. These are well known markers for the cancer immunology community and we feel that the description of these markers in the Introduction section would compromise its quality.

-Statistical analysis is poorly described. Please supplement with the specific tests used to evaluate the results. Justify Pearson tests used for linear correlations.

The last version of the manuscript does no longer make use of Pearson but Spearman correlation as the data is essentially non-parametric (once again we wonder whether the previous point-by-point answers were provided to the Reviewer). Also, specific tests have been referred to in all figures or text sections where P values are highlighted.

-Please avoid discussion in the results section since you have already a discussion section.

We have transferred all discussion-related content to the discussion section unless we found it necessary to provide some context to specific investigations.

-Which kind of survivals have you analysed? PFS or OS?

In the material methods and results section we now extensively refer to overall survival. (already addressed in previous version of the manuscript and reply to the Reviewers.

Reviewer 2 Report

The manuscript has been improved and can be accepted in the present form.

Author Response

We thank the Reviewer for their suggestion that improved the manuscript.

Round 2

Reviewer 1 Report

Thanks for the revised version of the manuscript

This manuscript is a resubmission of an earlier submission. The following is a list of the peer review reports and author responses from that submission.

Round 1

Reviewer 1 Report

The article entitled: A paradoxical role for regulatory T cells in the tumor microen- 2 vironment of pancreatic cancer; by Brouwer et al., is overall well written; however, it needs some amendments to improve the manuscript before publication. Tumor microenvironment is not only composed by SEVERAL cell populations by also by a COMPLEX extracellular matrix which make PDAC a high desmoplasic tumor.  Please find below my comments:

1. Introduction describes the role of some T-cell populations; however, it lacks information related to markers associated to the population you identified in the tumors e.g.: CD3, CD8, FOXP3, and CD163.

2. Statistical analysis is poorly described. Please supplement with the specific tests used to evaluate the results. Justify parametric tests used for linear correlations.

3. Figures show a poor resolution and figure legends are copy-pasted.

4. Please avoid discussion in the results section since you have already a discussion section. Please focus on results description.

5. Which kind of survival have you analysed PFS or OS?

6. It is quite interesting how you did not find any correlation/association with Treg infiltration in none of the specific transcriptional profiles of PDAC since one of the molecular subtypes proposed by Bailey et al., the immunogenic subtype, is characterized by T and B cell infiltration in tumors and FOXP3 mRNA expression. Discussion concerning this point is also poor.

Reviewer 2 Report

Brouwer et al. provide a concise and interesting article on the role of Tregs in the TME of pancreatic cancer. The clinical relevance of this study is obvious, since Tregs are shown to exhibit a protective role and to be associated with an improved overall survival in PDAC patiens. This can be used as a starting point for further research.

The introduction is well developed, the article incl. the results and discussion part is short but precise.

Major points:

Figure 1: at least to me the quality of the figure panel seems to be diminished/ needs to be improved

Discussion: Please point out limitations of your study and also possible future perspectives; clinical/ translational significance and possibilites as mentioned in the abstract and conclusion.

Minor points:

Line 43, 149: Empty space missing

Line 86: Formalin-fixed, paraffin-embedded (…)

Line 110: labeled vs. labelled (line 112,113), be consistent

Line 223: “mouse” models

Line 226, 232, 291, 293 (…): beta symbol

Line 257-258: Sentence needs to be corrected (“are present often present (…)”)              

Reviewer 3 Report

The article by Brouwer et al described a paradoxical role of Tregs in the prognosis of pancreatic cancer patients. They analyzed 122 PDAC tissue samples and performed multispectral immunofluorescence analyses to identify different infiltrating immune cell populations in the tumor and stroma. Surprisingly, they observed that higher Treg infiltration was associated with improved patient survival but CD+ T cell infiltration did not have any impact on overall survival. This observation is contrary to the generally accepted notion that higher CD8+ T cell infiltration is good for inhibition of tumor growth whereas higher Treg infiltration is typically correlated with worse outcome. However, there are few previous studies that is supportive of the data presented here, which suggest that there is more complexity in the tumor immunity than typically realized. Hence, this study is certainly of interest to cancer research community and warrant further investigations to delineate the role of Tregs in pancreatic cancer.